# Robustness to Unbounded Smoothness of Generalized SignSGD

**Michael Crawshaw**[*]
George Mason University
mcrawsha@gmu.edu

**Mingrui Liu**[*]
George Mason University
mingruil@gmu.edu

**Francesco Orabona**[*]
Boston University
francesco@orabona.com

**Wei Zhang**[*]
IBM T. J. Watson Research Center
weiz@us.ibm.com

**Zhenxun Zhuang**[*]
Meta Platforms, Inc.
oldboymls@gmail.com

## Abstract

Traditional analyses in non-convex optimization typically rely on the smoothness assumption, namely requiring the gradients to be Lipschitz. However, recent evidence shows that this smoothness condition does not capture the properties of some deep learning objective functions, including the ones involving Recurrent Neural Networks and LSTMs. Instead, they satisfy a much more relaxed condition, with potentially unbounded smoothness. Under this relaxed assumption, it has been theoretically and empirically shown that the gradient-clipped SGD has an advantage over the vanilla one. In this paper, we show that clipping is not indispensable for Adam-type algorithms in tackling such scenarios: we theoretically prove that a generalized SignSGD algorithm can obtain similar convergence rates as SGD with clipping but does not need explicit clipping at all. This family of algorithms on one end recovers SignSGD and on the other end closely resembles the popular Adam algorithm. Our analysis underlines the critical role that momentum plays in analyzing SignSGD-type and Adam-type algorithms: it not only reduces the effects of noise, thus removing the need for large mini-batch in previous analyses of SignSGD-type algorithms, but it also substantially reduces the effects of unbounded smoothness and gradient norms. To the best of our knowledge, this work is the first one showing the benefit of Adam-type algorithms compared with non-adaptive gradient algorithms such as gradient descent in the unbounded smoothness setting. We also compare these algorithms with popular optimizers on a set of deep learning tasks, observing that we can match the performance of Adam while beating others.

## 1 Introduction

Recent years have witnessed a surge in non-convex machine learning models, with a focus on deep neural networks [27]. DNNs have achieved tremendous progress in a variety of tasks, including computer vision [26, 18, 23], natural language processing [11, 50], and a lot more. Despite their huge empirical success, the theoretical analyses of non-convex optimization [21] prove to be fundamentally more challenging than the established convex optimization theory [4]. Among the numerous literature, many of them assume smoothness of the objective function, namely requiring the gradients to be Lipschitz. Under this scenario, past works have succeeded in proving the convergence rates for a number of algorithms, e.g., Stochastic Gradient Descent [14], AdaGrad [52, 30], and STORM [9, 7].

Nevertheless, it was recently observed that the smoothness assumption does not capture the training of LSTMs [20]: the Hessian can grow with the size of the gradients [56]. Inspired by this, Zhang et

---

*Authors in alphabetical order.

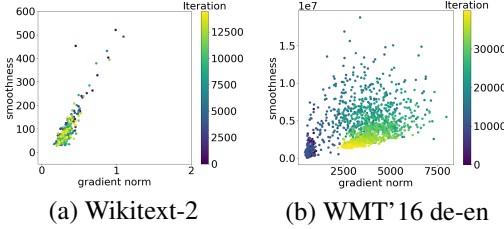

(a) Wikitext-2     (b) WMT'16 de-en

Figure 1: Local gradient Lipschitz constant vs. Gradient norm on training (a) a 2-layer Transformer Encoder on Wikitext-2 (b) a 6-layer Transformer on WMT'16 Multimodal Machine Translation de-en dataset. The colorbar indicates #Iterations in training. Details in Section 5.2.

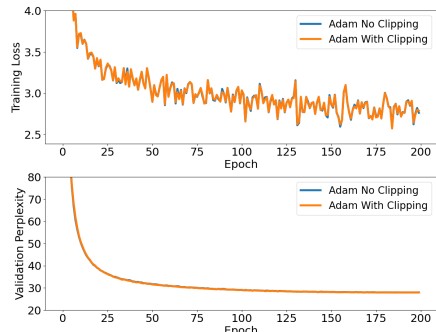

Figure 2: Training GPT-2 on Wikitext-103 using Adam with or without gradient clipping.

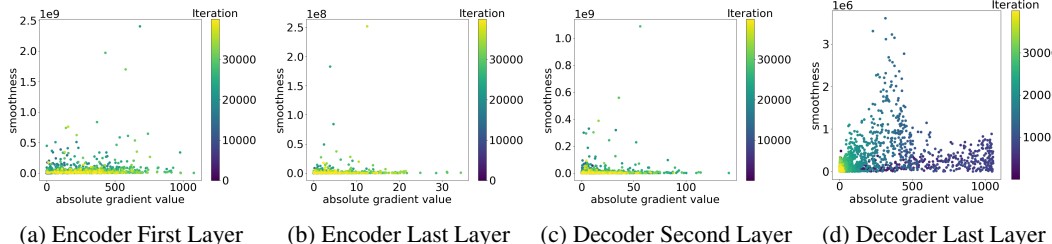

(a) Encoder First Layer     (b) Encoder Last Layer     (c) Decoder Second Layer     (d) Decoder Last Layer

Figure 3: Local gradient Lipschitz constant vs. absolute gradient value on training a Transformer on WMT'16 Multimodal Translation de-en dataset. Each figure represents a randomly picked coordinate in corresponding layers. The colorbar indicates #Iterations during training. Details in Section 5.2.

al. [56] proposed a relaxed smoothness assumption, named $(L_0, L_1)$ smoothness:

$$\|\nabla^2 F(\boldsymbol{x})\| \leq L_0 + L_1 \|\nabla F(\boldsymbol{x})\| . \tag{1}$$

They also showed that the well-known gradient clipping technique can ensure Stochastic Gradient Descent's (SGD) convergence in such scenarios. Later, their results were improved to show that SGD with clipping can be made unaffected by the $L_1$ in (1) and is able to recover the optimal convergence rate of SGD under the original smoothness setting [55, 22].

Nevertheless, the $(L_0, L_1)$ condition has not yet been empirically verified beyond LSTMs. Therefore, our **first contribution** lies in studying the applicability and generalization of the $(L_0, L_1)$ condition. In particular, we have empirically verified that the popular Transformer [50] model also seems to satisfy this assumption, see Figure 1. Yet, we noticed that different coordinates, especially when they are in different layers of the model, exhibit very distinct $L_0$ and $L_1$ values as shown in Figure 3. Hence, we propose to refine the $(L_0, L_1)$ assumption in (1) to a coordinate-wise version (Assumption 2) and consider this to better capture the loss surface when training deep neural networks like Transformers.

Given that we assume (a generalization) of the $(L_0, L_1)$ assumption, it would be natural to use some clipping procedure. However, we found out that the use of clipping on Adam [25], while carried out in common practice [e.g., 53], *has no effect on the training and testing performance on optimizing a large transformer model as shown in Figure 2*. In retrospect, this might not be surprising: It is known that Adam has an implicit clipping behavior due to the normalization by the estimated second moment of the gradients. Indeed, Adam can be interpreted as a variant of SignSGD [2].

Inspired by this, our **second contribution** is to propose and analyze a *generalized SignSGD* algorithm under the relaxed smoothness assumption. It is parameterized in such a way that it on one end recovers SignSGD while on the other end closely resembles Adam. Apart from the convergence rates, we also located the critical role the momentum plays in analyzing Adam-type algorithms: it not only reduces the effects of noise but also gives an exponential decaying effect on the unbounded gradient norms and smoothness. This can partly explain the phenomenon that clipping does not help Adam. Moreover, we show a gap between the upper bound of SignSGD and the lower bound of SGD

in the relaxed smoothness setting. This can be considered as a first step in explaining the superior performance of Adam in practical deep learning problems.

The structure of this paper is as follows. Section 2 discusses related works and how our paper builds upon and distinguishes from them. The settings and assumptions are carried out in Section 3. We will introduce formally the generalized SignSGD algorithm and its analysis in Section 4, with a detailed discussion on the bounds and the role of momentum. The experimental results are shown in Section 5, comparing our algorithm with some popular competitors in deep learning tasks. Finally, we draw some conclusions and discuss the limitations of our work in Section 6.

**Notations** We will use $[d]$ to denote the sequence $[1, 2, \ldots, d]$ and use bold letters to represent vectors, e.g., $\boldsymbol{u} \in \mathbb{R}^d$. The $j$-th coordinate of a vector $\boldsymbol{u}$ is $u_j$. Throughout this paper, we study the Euclidean space $\mathbb{R}^d$ with the inner product $\langle \cdot, \cdot \rangle$. $\mathbb{E}[\boldsymbol{u}]$ means the expectation with respect to the underlying probability distribution of a random variable $\boldsymbol{u}$, and $\mathbb{E}_t[\boldsymbol{u}]$ is the conditional expectation of $\boldsymbol{u}$ conditioned on the past of time $t$. The gradient of $F$ at $\boldsymbol{x}$ is denoted by $\nabla F(\boldsymbol{x})$. We use $\mathbb{I}(\cdot)$ to denote the indicator function, $\|\boldsymbol{u}\|_p$ to denote the $p$-norm: $\|\boldsymbol{u}\|_p := (\sum_{j=1}^d |u_j|^p)^{1/p}$ and $\|\boldsymbol{u}\|_\infty$ the maximum norm: $\|\boldsymbol{u}\|_\infty := \max\{|u_1|, \ldots, |u_d|\}$. We also denote by $\sum_{k=i}^j x_k = 0$ when $i > j$.

## 2 Related Works

**Adaptive Gradient Methods** Adaptive gradient methods [34, 12, 25, 19, 44] are popular optimizers for training deep neural networks. The traditional analysis of adaptive gradient methods is providing regret bounds under the online convex optimization framework [12, 25, 44]. Recently, there are some analysis of adaptive gradient methods for nonconvex smooth functions [6, 5, 54, 10, 61]. Zou et al. [60] introduces an intriguing connection between Adam [25] and SignGD [3] when training a two-layer neural network in the deterministic setting, where SignGD is an algorithm following the negative gradient sign direction to perform the update. However, these works cannot be directly extended to nonconvex functions with unbounded smoothness in the stochastic setting. To the best of our knowledge, this work is the first one establishing guarantees for coordinate-wise type optimizers like generalized SignSGD as well as Adam-type updates under a relaxed smoothness condition.

**Gradient Clipping** The algorithm and analysis of gradient clipping can be traced back to [1, 48, 13] under the assumption that the function is convex and rapidly growing. Hazan et al. [17] considered gradient clipping in quasi-convex optimization. Mai and Johansson [32] showed the stability and convergence of stochastic gradient clipping algorithms for convex problems without the smoothness condition. Gradient clipping is a standard technique in training deep neural networks [39, 40] such as RNNs and LSTMs. The theoretical analysis of gradient clipping for nonconvex models is pioneered by [56], in which the authors analyzed the convergence of gradient clipping under the relaxed smoothness assumption rather than the standard smoothness assumption. Zhang et al. [55] further improved the convergence rate bound under the same assumption as in [56]. Gradient clipping is also used when there is a heavy tail noise in the stochastic gradient to establish high probability convergence rates [8, 15, 57]. Cutkosky and Mehta [7] proved that normalized momentum improves normalized SGD under a second-order smoothness condition. A close algorithm is the one in [22] which employs gradient normalization, momentum, and no gradient clipping to tackle the $(L_0, L_1)$ condition (1) and control noise. Yet, their algorithm normalizes each coordinate with the same scale unlike popular optimizers such as Adam [25]. Moreover, we observe empirically that normalized SGD with momentum performs worse than Adam. Motivated by this, we propose a coordinate-wise optimization algorithm which requires new analysis tools compared with [22].

**Employ $m_t^2$ to compute $v_t$ in Adam** Designed to combine the advantages of Adagrad [12] and RMSProp [49], the update of Adam [25] employs the ratio between the exponential moving average of the stochastic gradient ($\boldsymbol{m}_t$) and the exponential moving average of the squared stochastic gradient ($\boldsymbol{v}_t$). Many variants of Adam have been proposed ever since. Among them, one idea is to use $\boldsymbol{m}_t^2$ to compute $\boldsymbol{v}_t$ instead of $\boldsymbol{g}_t^2$. The intuition is that $\boldsymbol{m}_t$ represents a better update direction than $\boldsymbol{g}_t$ and can thus better capture the second-moment information. Reddi et al. [43] adopted this change to prove the convergence of Adam in a federated learning setting; yet, they only consider the smooth setting and require a large $\epsilon$ to obtain convergence in contrast to the original Adam. Later, Wang et al. [51] explored this idea in more detail, but their analyses are still restricted to the smooth setting. There also exist other variants of the Adam update that attempt to obtain a more stable update changing the order of the normalization and momentum operations [see, e.g., 59].

# 3 Settings and Preliminaries

In this paper, we focus on the following stochastic optimization problem:

$$\min_{\boldsymbol{x}\in\mathbb{R}^d} F(x) := \mathbb{E}_{\xi\sim\mathcal{D}}[f(\boldsymbol{x},\xi)],$$

where $\xi$ is a random variable representing a randomly selected data sample or random noise following an unknown distribution $\mathcal{D}$. We will use the following assumptions.

**Assumption 1.** $F : \mathbb{R}^d \to \mathbb{R}$ *is differentiable and bounded from below with infimum* $F^*$.

**Assumption 2.** *We say that a differentiable function $F(\boldsymbol{x})$ is $(\boldsymbol{L_0}, \boldsymbol{L_1})$-smooth coordinate-wisely, if for any $\boldsymbol{x}, \boldsymbol{y} \in \mathbb{R}^d$ for which $\|\boldsymbol{x} - \boldsymbol{y}\|_2 \leq \frac{1}{\|\boldsymbol{L_1}\|_\infty}$, we have for any $j \in [d]$ that*

$$\left| \frac{\partial F}{\partial x_j}(\boldsymbol{y}) - \frac{\partial F}{\partial x_j}(\boldsymbol{x}) \right| \leq \left( \frac{L_{0,j}}{\sqrt{d}} + L_{1,j} \left| \frac{\partial F}{\partial x_j}(\boldsymbol{x}) \right| \right) \|\boldsymbol{y} - \boldsymbol{x}\|_2 . \tag{2}$$

*We will denote $\boldsymbol{L_0} := [L_{0,1}, L_{0,2}, \ldots, L_{0,d}]^T$ and $\boldsymbol{L_1} := [L_{1,1}, L_{1,2}, \ldots, L_{1,d}]^T$.*

The original $(L_0, L_1)$ smoothness assumption (1) in [56] was proposed as a generalization of the more common smoothness assumption, which says that the gradient should be Lipschitz. Indeed, when $L_{1,j}$ are zero, we recover the smoothness assumption. In contrast, when $L_{1,j}$ are non-zero, the smoothness of the function is potentially *unbounded*. Yet, [56] works with norms and applies to the global scale, while ours is more fine-grained and applies to each coordinate separately. One motivation for this assumption comes from [Remark 2.3, 55] where they noted that (1) can be relaxed to an assumption on gradient differences: there exists $K_0, K_1 > 0$ such that

$$\|\nabla F(\boldsymbol{x}) - \nabla F(\boldsymbol{y})\|_2 \leq (K_0 + K_1 \|\nabla F(\boldsymbol{x})\|_2)\|\boldsymbol{x} - \boldsymbol{y}\|_2, \ \forall \boldsymbol{x}, \boldsymbol{y} \in \mathbb{R}^d : \|\boldsymbol{x} - \boldsymbol{y}\|_2 \leq 1/K_1 . \tag{3}$$

Indeed, our Assumption 2 implies (3) when $L_{0,j} = L_0$ and $L_{1,j} = L_1$ for all $j \in [d]$, up to constants (See Lemma 3 in the Appendix). Note that the $\frac{1}{\sqrt{d}}$ factor in ours is exactly for easy comparison with (3). The reason we turn to the current coordinate-wise version is that we observed a vast variance across different layers in training Transformer models: (1) is still true globally (Figure 1), but each layer or even each coordinate satisfies has a very different $(L_0, L_1)$ pair (Figure 3). The smoothness assumption has been generalized in orthogonal directions in other work [45, 3, 24].

One merit of Assumption 2 is that it gives us the following descent lemma.

**Lemma 1.** *Let $F$ be $(\boldsymbol{L_0}, \boldsymbol{L_1})$-smooth coordinate-wisely. Then, for any $\boldsymbol{x}, \boldsymbol{y} \in \mathbb{R}^d$ for which $\|\boldsymbol{x} - \boldsymbol{y}\|_2 \leq \frac{1}{\|\boldsymbol{L_1}\|_\infty}$, we have*

$$F(\boldsymbol{y}) \leq F(\boldsymbol{x}) + \langle \nabla F(\boldsymbol{x}), \boldsymbol{y} - \boldsymbol{x} \rangle + \sum_{j=1}^{d} \frac{1}{2} \left( \frac{L_{0,j}}{\sqrt{d}} + L_{1,j} \left| \frac{\partial F}{\partial x_j}(\boldsymbol{x}) \right| \right) \|\boldsymbol{y} - \boldsymbol{x}\|_2 |y_j - x_j| .$$

Our last assumption is common in the literature studying the $(L_0, L_1)$ smooth condition [56, 55].

**Assumption 3.** *For each $j \in [d]$, there exists $\sigma_j > 0$ such that for all $\boldsymbol{x} \in \mathbb{R}^d$ and $\xi \sim \mathcal{D}$, the noise satisfies $\left| [\nabla f(\boldsymbol{x}, \xi)]_j - \frac{\partial F}{\partial x_j}(\boldsymbol{x}) \right| \leq \sigma_j$ with probability 1. We will denote $\boldsymbol{\sigma} := [\sigma_1, \sigma_2, \ldots, \sigma_d]^T$.*

# 4 A Generalized SignSGD Algorithm

---
**Algorithm 1** Generalized SignSGD *(All operations on vectors are element-wise.)*

---
1: Inputs: $\boldsymbol{x}_1, \beta_1, \beta_2, \eta$
2: $\boldsymbol{m}_0 = 0, \boldsymbol{v}_0 = 0$
3: **for** $t = 1, \cdots, T$ **do**
4:     Compute an unbiased estimate $\nabla f(\boldsymbol{x}_t, \xi_t)$ of $\nabla F(\boldsymbol{x}_t)$, denoted as $\boldsymbol{g}_t$
5:     $\boldsymbol{m}_t = \beta_1 \boldsymbol{m}_{t-1} + (1 - \beta_1)\boldsymbol{g}_t$
6:     $\boldsymbol{v}_t = \beta_2 \boldsymbol{v}_{t-1} + (1 - \beta_2)\boldsymbol{m}_t^2$
7:     $\boldsymbol{x}_{t+1} = \boldsymbol{x}_t - \eta \frac{\boldsymbol{m}_t}{\sqrt{\boldsymbol{v}_t}}$
8: **end for**

---

In this section, we present in Algorithm 1 a generalized SignSGD algorithm. This algorithm encompasses a variety of optimization algorithms.

At first sight, it seems very similar to Adam. Indeed, if we employ $g_t^2$ in computing $v_t$ instead of $m_t^2$, then it is exactly Adam, except for the bias correction terms. We would like to clarify that the idea of this change has been explored before, as detailed in Section 2. In this paper, the motivation for adopting this idea comes from the known effect of momentum on reducing the influence of noises [7]. Indeed, in our analysis the difference between $m_t$ and $\nabla F(x_t)$ is much more controllable than between $g_t$ and $\nabla F(x_t)$. Thus, we consider employing $m_t$ in computing $v_t$ a better choice.

On the other end, the careful reader might observe that Algorithm 1 recovers the SignSGD with Momentum algorithm, also called SIGNUM in [3], when setting $\beta_2 = 0$. Sign-based algorithms are naturally suited to distributed learning [29] and the idea dated back to at least RPROP [46]. The convergence to a stationary point (with $\ell_1$ norm) under a coordinate-wise smoothness condition has been established for SignSGD with/without the momentum in [3] though they necessitate large mini-batches to control the variance of the noise. Yet, we are more interested in their property of the update size being bounded without the need for explicit clipping.

Note that both SignSGD and Adam are good candidates for optimization algorithms whose update must be bounded on functions that satisfy the $(L_0, L_1)$ condition. Indeed, SignSGD can be seen as an extreme form of gradient clipping. On the other hand, as said in the introduction, Adam does not seem to require gradient clipping at all when used to train the large Transformer model in Figure 2.

Hence, we expect our algorithm, a generalization of SignSGD and a close resemblance to Adam, can enjoy the merits of both and be robust to the unbounded smoothness in the $(L_0, L_1)$ scenario. In the next section, we will formalize this claim by presenting the theoretical analysis of Algorithm 1.

### 4.1 Theoretical Convergence Analysis

**Theorem 1.** *Under Assumptions 1, 2, and 3, assume $M_j := \sup\left\{\left|\frac{\partial F}{\partial x_j}(x)\right| : F(x) \le F(x_1)\right\}$ is finite for each $j \in [d]$, let $\Delta$ be any upper bound on $F(x_1) - F^*$, $\alpha = \min\left(\frac{\sqrt{\|L_0\|_1}\sqrt{\Delta}}{\|\sigma\|_1\sqrt{T}}, 1\right)$, $\beta_1 = 1 - \alpha$, $\frac{\sqrt{\beta_2}}{\beta_1} < 1$, $\rho = 1 - \frac{\sqrt{\beta_2}}{\beta_1}$, $\eta = \frac{\sqrt{\Delta\alpha}}{\sqrt{\|L_0\|_1}\sqrt{T}}$, for $T \ge \max\left(\frac{100d\Delta\|L_1\|_\infty^2}{(1-\beta_2)\rho^2\|L_0\|_1}, \frac{10000d^2\Delta\|\sigma\|_1^2\|L_1\|_\infty^4}{(1-\beta_2)^2\rho^4\|L_0\|_1^3}\right)$, Algorithm 1 guarantees, with probability at least $1 - \delta$, that*

$$\min_{t\in[T]} \|\nabla F(x_t)\|_1 = \mathcal{O}\left(\frac{\sqrt{\log(dT/\delta)}\|L_0\|_1^{1/4}\Delta^{1/4}\|\sigma\|_1^{1/2}}{\rho\sqrt{1-\beta_2}T^{1/4}} + \frac{\log(dT/\delta)\sqrt{\|L_0\|_1\Delta}}{\rho\sqrt{T}}\right)$$
$$+ \mathcal{O}\left(\frac{\|M\|_1 + \|\sigma\|_1}{\rho}\exp\left(-\frac{\sqrt{1-\beta_2}\|L_0\|_1^{3/4}}{\sqrt{d}\|L_1\|_\infty\|\sigma\|_1^{1/2}\Delta^{1/4}}T^{1/4}\right) + \frac{\|\nabla F(x_1)\|_1}{T}\right).$$

*Furthermore, for the case when $\beta_2 = 0$, we have the following refined guarantee:*

$$\min_{t\in[T]} \|\nabla F(x_t)\|_1 = \mathcal{O}\left(\frac{\sqrt{\log(dT/\delta)}\|L_0\|_1^{1/4}\Delta^{1/4}\|\sigma\|_1^{1/2}}{T^{1/4}} + \frac{\log(dT/\delta)\sqrt{\|L_0\|_1\Delta}}{\sqrt{T}}\right)$$
$$+ \mathcal{O}\left(\frac{\|\nabla F(x_1)\|_1}{\sqrt{T}}\left(\frac{1}{\sqrt{T}} + \frac{\|\sigma\|_1}{\sqrt{\|L_0\|_1\Delta}}\right) + \frac{\|\sigma\|_1}{T}\right).$$

Here, $M_j$ denotes the maximum absolute value of the partial derivative of $F$ for coordinate $j$ among the sub-level set of $F(x_1)$, namely any point $x$ with $F(x) \le F(x_1)$. In other words, we assume gradients to be bounded in the sub-level set of $F(x_1)$; yet, we do not make any restriction on gradients outside of this set. We believe this is not a strong assumption, for example, when the sub-level set of $F(x_1)$ is bounded, then by the assumed continuity of gradients it trivially holds. Also, we just require an upper bound and it can even be exponentially large as we have an exponentially decaying coefficient to counteract it: notice how the term $\|M\|_1$ is multiplied by a term that decays exponentially with $T$. Better still, when $\beta_2 = 0$, we no longer even need this assumption and the algorithm is entirely free of the influence of $\|M\|_1$. To see why this is good, we show a refined lower bound of Gradient Descent under the relaxed smoothness scenario below which is originally in [56].

**Theorem 2.** *Fix $\epsilon > 0, L_0 > 0, L_1 > 0, M \geq \max(\frac{L_0}{L_1}, \epsilon)$, and $x_0 \in \mathbb{R}$. Pick any constant learning rate $\eta$ for GD, with the knowledge of the above constants. Then, there exists a 1-d $(L_0, L_1)$-smooth function, bounded from below by $f^*$ (finite), and such that $\sup\{|f'(x)| : f(x) \leq f(x_0)\} \leq M$ on which the number of iterations $T$ of GD with learning rate $\eta$ to guarantee $|f'(x_T)| < \epsilon$ is at least*

$$\frac{ML_1(f(x_0) - f^* - \frac{15\epsilon^2}{16L_0})}{2\epsilon^2 \left(\ln \frac{ML_1}{L_0} + 1\right)} .$$

Theorem 2 shows that in the relaxed smoothness setting, GD with any constant step size will suffer from a linear term depending on $L_1 M$. On a side note, it is a fixed version of the lower bound in [56]: we provide in Appendix an explanation of errors in their lower bound and our corrected proof.

Compared with GD, our algorithm only has an exponentially decaying dependence on $L_1 M$. We consider this to be substantial merit of our algorithm. Furthermore, when $\beta_2 = 0$ in which case we recover the SignSGD with Momentum algorithm, we can even show that it completely removes the effects of the unbounded gradient norms. Also notice that in such case we actually no longer need the assumption of $M_j := \sup \left\{ \left| \frac{\partial F}{\partial x_j}(\boldsymbol{x}) \right| : F(\boldsymbol{x}) \leq F(\boldsymbol{x}_1) \right\}$ being finite for each $j \in [d]$ anymore, and the $\|\boldsymbol{L_1}\|_\infty$ term does not appear in the final bound anymore.

We also would like to point out that this bound closely resembles the one achieved by SGD with gradient clipping algorithm [55] except that we consider the coordinate-wise setting: take the setting of $\beta_2 = 0$ for example, we need at most $\mathcal{O}\left(\Delta \max\left\{ \frac{\|\boldsymbol{\sigma}\|_1^2\|\boldsymbol{L_0}\|_1}{\epsilon^4}, \frac{d^2\|\boldsymbol{\sigma}\|_1^2\|\boldsymbol{L_1}\|_\infty^4}{\|\boldsymbol{L_0}\|_1^3}, \frac{d\|\boldsymbol{L_1}\|_\infty^2}{\|\boldsymbol{L_0}\|_1} \right\}\right)$ to get a point $\boldsymbol{x}$ with $\|\nabla F(\boldsymbol{x})\|_1 \leq \epsilon$ with high probability.

**Remark 1** The almost surely bounded assumption 3 can be relaxed to sub-gaussian noise, using standard extensions of Freedman inequality [e.g., 16].

**Remark 2** When $\beta_2 = 0$, we can prove an average-iterate complexity bound (see Proof of Theorem 1 for $\beta_2 = 0$ in Appendix A.3); yet, we use the min form for consistency between the two cases.

**Remark 3** Our bound is incomparable with the one in [55, Theorem 3.2]. Yet, as we said, if $L_{0,j} = L_0$ and $L_{1,j} = L_1$ for all $j \in [d]$, then the function satisfies (3). In this case, assuming the noise vector and the gradient vector to be dense to be able to compare the $\ell_1$-norm and the $\ell_2$-norm, we recover the same bound of [55, Theorem 3.2] in terms of dependencies on $L_1, L_0$, and $T$. Instead, in the more general case when $L_{0,j}$ and $L_{1,j}$ are not uniform vectors, our bound allows a finer control of the unbounded smoothness.

**Remark 4** Careful readers might be concerned on the relations between $\alpha, \beta_1, \beta_2, \rho$, and $T$ when $\alpha \neq 1$. We would like to note that, when $\beta_2$ is fixed, $\alpha$ is inversely proportional to $\sqrt{T}$. In turn, the definition of $\rho$ means that as $T$ grows, $\rho$ grows and approaches $1 - \sqrt{\beta_2}$. Thus, the two conditions for $T$ decreases when $T$ grows. This means that there must exists a threshold of $T$ above which the two conditions on $T$ always hold. In summary, Theorem 1 conveys the same message as [55] that as long as the expected $\epsilon$ is sufficiently small, the complexity no longer has a dependency on $\boldsymbol{L_1}$.

The proof of the theorem is highly technical and it uses recent advancements in the analysis of momentum methods [7], key techniques to deal with the $(L_0, L_1)$ assumption [55], as well as a novel and essential inductive argument to control the norm of past gradients. We want to stress that the difficulty mainly comes from analyzing Adam-type updates when $\beta_2 > 0$, while for the other case of $\beta_2 = 0$ the proof is significantly simpler. The full proof is in the Appendix, but here we present a proof sketch that underlines the main steps. First, we list some key lemmas we used but move their proofs to the appendix due to space constraints.

**Lemma 5.** *With notations in Algorithm 1, for $\tau \leq \bar{\tau} = \frac{\sqrt{1-\beta_2}}{\eta\sqrt{d}\|\boldsymbol{L_1}\|_\infty}$, we have $\|\boldsymbol{x}_{t-\tau} - \boldsymbol{x}_t\|_2 \leq \frac{1}{\|\boldsymbol{L_1}\|_\infty}$.*

Lemma 5 limits our focus to the most recent $\bar{\tau}$ steps on which Assumption 2 and Lemma 1 can apply.

**Lemma 7.** *Assume Assumption 3. With the notation of Algorithm 1, let $j \in [d]$ and $\beta_1 \leq 1$. Then, with probability at least $1 - 3\delta$, for any $t_0 \in [t]$, we have*

$$\left| \sum_{\tau=1}^{t_0} \beta_1^{t-\tau}\left(g_{\tau,j} - \frac{\partial F}{\partial x_j}(\boldsymbol{x}_\tau)\right) \right| \leq 3\sigma_j \max(1, \log(1/\delta)) + \frac{3}{\sqrt{1-\beta_1^2}}\sqrt{\sigma_j^2 \max(1, \log(1/\delta))} \triangleq E_j .$$

Lemma 7 is the major tool we use to handle the noise we incur during drawing stochastic gradients. It is derived based on Lemma 12 in [8].

**Lemma 10.** *With the notation of Algorithm 1 and under the assumptions of Theorem 1, if $\left|\frac{\partial F}{\partial x_j}(\boldsymbol{x}_\tau)\right| \leq M_j$ holds for all $\tau \leq t$ and $j \in [d]$, and $D > 0$, then, with probability at least $1 - 3t\delta$ we have that,*

$$\text{either } \left|\frac{\partial F}{\partial x_j}(\boldsymbol{x}_t)\right| < \frac{5B_j}{D} \text{ or } \frac{|m_{t,j}|}{\sqrt{v_{t,j}}} \geq \frac{\rho D}{5\sqrt{1-\beta_2}} \,,$$

*where $B_j \triangleq \frac{\eta L_{0,j}}{\sqrt{1-\beta_2}(1-\beta_1)} + \beta_1^{\bar{\tau}}(M_j + \sigma_j) + (1-\beta_1)E_j$ and $D \triangleq 1 - \frac{2\eta\sqrt{d}\|\boldsymbol{L_1}\|_\infty}{\sqrt{1-\beta_2}(1-\beta_1)}$.*

Lemma 10 is similar to Lemma A.2 in [60] which considered the deterministic and smooth setting; in contrast, our proof is much more challenging in that we need to tackle both the noise and the unbounded smoothness. With this lemma, we know that either the true gradient is small or that the update of our Algorithm 1 can be lower bounded.

**Lemma 12.** *Under Assumptions 1, 2, and 3, using the hyperparameters in Theorem 1, denoting $\alpha = 1 - \beta_1$ and $\boldsymbol{\epsilon}_t = \boldsymbol{m}_t - \nabla F(\boldsymbol{x}_t)$, for all $t$ and $j \in [d]$ we have, with probability at least $1 - 3\delta$,*

$$|\epsilon_{t+1,j}| \leq (1-\alpha)^t \left(\alpha\sigma_j + (1-\alpha)\left|\frac{\partial F}{\partial x_j}(\boldsymbol{x}_1)\right|\right) + \frac{\eta L_{0,j}}{\alpha} + \alpha E_j + (1-\alpha)\eta\sqrt{d}L_{1,j}\sum_{\tau=0}^{t-1}(1-\alpha)^\tau\left|\frac{\partial F}{\partial x_j}(\boldsymbol{x}_{t-\tau})\right| \,.$$

Lemma 12 shows how the use of momentum can help control the noise by choosing $\beta_1$ wisely. It is adapted from the proof of Theorem 2 in [8] but with the added difficulty of unbounded smoothness.

*Proof sketch of Theorem 1.* Observing the formula of setting $\beta_1$, we can see that when $\|\boldsymbol{\sigma}\|_1 \leq \sqrt{\|\boldsymbol{L_0}\|_1\Delta}/\sqrt{T}$, $\beta_1 = 0$. As $\beta_2 < \beta_1$, Algorithm 1 reduces to SignSGD. In this case, the key component is Lemma 12 using which we are able to show that $\sum_{t=1}^T\left|m_{t,j} - \frac{\partial F}{\partial x_j}(\boldsymbol{x}_t)\right|$ can be controlled as $C_1\sum_{t=1}^T\left|\frac{\partial F}{\partial x_j}(\boldsymbol{x}_t)\right| + C_2$. The summation of true gradients over time can then be offsetted by choosing $\eta$ and $\beta_1$ wisely when we invoke the descent lemma 1. The rest is standard.

Now for the other case in which $\|\boldsymbol{\sigma}\|_1 > \sqrt{\|\boldsymbol{L_0}\|_1\Delta}/\sqrt{T}$, we take a different route.

First, notice that Assumption 2 and the Descent Lemma 1 only hold when two points are not too far away. Thus, we need to restrict our attention to the recent updates (Lemma 5), beyond which we would have no control. This means we want the influence of those updates too long ago to not have a big effect on the current one. To make this happen, one natural idea is to use a bounded gradient assumption, then with the use of exponential averaging, their effect would be quickly reduced. Yet, assuming directly that all gradients are bounded would trivialize the $(\boldsymbol{L_0}, \boldsymbol{L_1})$ assumption. Thus, we pose a much weaker condition, assuming that $M_j := \sup\left\{\left|\frac{\partial F}{\partial x_j}(\boldsymbol{x})\right| : F(\boldsymbol{x}) \leq F(\boldsymbol{x}_1)\right\}$ being finite for each $j \in [d]$. Then, we prove that $M_j$ will provide an upper bound to all the true gradients the algorithm see. We prove it using induction, analyzing separately the case that either the true gradient is already very small and we have reached the proximity of a stationary point, or the objective function is monotonically non-increasing and the gradient remains bounded.

Having controlled the past gradients, we prove in Lemma 10 that the update of Algorithm 1 is either very small that we can pass or having a constant lower bound that we can use in the Descent Lemma 1.

Also, considering that this is the stochastic setting, noise typically slows down convergence or can even cause the algorithm to diverge if the hyperparameters are not chosen wisely. To handle this, we invoke Freedman's inequality to show that the addition of adjacent stochastic noise almost cancels out each other and the absolute value of the sum remains controlled (Lemma 7).

Yet, we still need another block to handle the difference between the true gradient and the momentum as we are updating in the direction of the momentum instead of the true gradient. Turns our that we can prove that $\text{sign}(m_{t,j}) = \text{sign}\left(\frac{\partial F}{\partial x_j}(\boldsymbol{x}_t)\right)$ when $\left|\frac{\partial F}{\partial x_j}(\boldsymbol{x}_t)\right|$ is not too small. As before, in the case $\left|\frac{\partial F}{\partial x_j}(\boldsymbol{x}_t)\right|$ is small, we have converged on that coordinate. Combining all these blocks together, we are able to arrive at the final results. $\square$

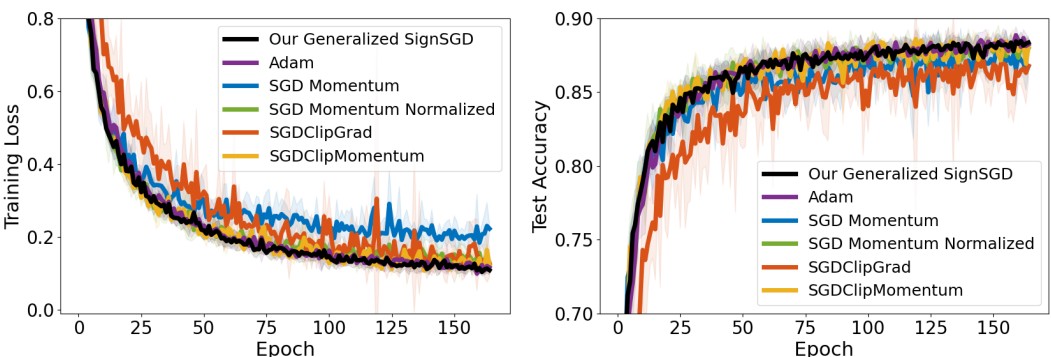

Figure 4: Training a 20-layer Resnet on CIFAR10. The shading of each curve represents the 95% confidence interval computed across 5 independent runs from different random seeds.

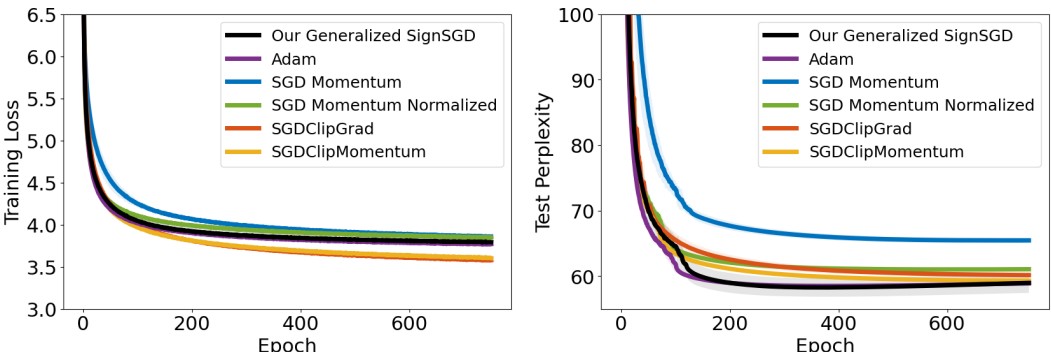

Figure 5: Training an AWD-LSTM to do language modeling (word level) on Penn Treebank. The shading of each curve represents the 95% confidence interval over 5 independent runs.

## 5   Experiments

We conducted our experiments using PyTorch [41] on Nvidia V100 GPUs. Codes can be found at
`https://github.com/zhenxun-zhuang/Generalized-SignSGD`.

### 5.1   Comparison with Other Optimizers

To validate the efficacy of our Algorithm 1, we compare it with Adam [25], SGD [47], SGD Momentum Normalized [22], SGDClipGrad, and SGDClipMomentum. The latter two are from Algorithm 1 in [55] where SGDClipGrad corresponds to the case when $\nu = 0$ and SGDClipMomentum corresponds to when $\nu = 1$.

**Training** Unless otherwise specified, we use grid-search to fine-tune the initial learning rate for all optimizers, as well as the clipping threshold for SGDClipGrad and SGDClipMomentum, and $\beta_2$ for Adam and our algorithm, to select the one giving the best validation performance on a separated validation set. We then employ the best performing hyperparameters to train the model over all training data and report the testing performance. The testing is repeated with random seeds 5 times to eliminate the influence of stochasticity. For more details, please refer to Section A.4.

**Resnet for Image Classification on CIFAR-10** We employ the 20-layer Residual Network model [18] to do image classification on the CIFAR-10 dataset. Images are normalized per channel using the means and standard deviations computed from all training images. We adopt the data augmentation technique following [28] (for training only): 4 pixels are padded on each side of an image and a 32 × 32 crop is randomly sampled from the padded image or its horizontal flip. The mini-batch size is 128 and we train all algorithms for 164 epochs. We do not employ any learning rate decay schedule in order to focus on the comparison of the optimizers themselves. We fixed the weight decay value to be

Table 1: Average final training loss and test accuracy achieved by each method when optimizing respective models on each dataset. The $\pm$ shows $95\%$ confidence intervals of the mean loss/accuracy/perplexity value over 5 runs starting from different random seeds.

| Methods | CIFAR10 | | Penn Treebank | |
|---|---|---|---|---|
| | Training loss | Test accuracy | Training loss | Test perplexity |
| SGD Momentum | $0.2226 \pm 0.0169$ | $0.8674 \pm 0.0048$ | $3.8587 \pm 0.0058$ | $65.4622 \pm 0.3842$ |
| SGD Momentum Normalized | $0.1262 \pm 0.0170$ | $0.8795 \pm 0.0086$ | $3.8487 \pm 0.0073$ | $61.0558 \pm 0.3224$ |
| SGDClipGrad | $0.1288 \pm 0.0403$ | $0.8677 \pm 0.0106$ | $\mathbf{3.5774 \pm 0.0081}$ | $60.1604 \pm 0.2797$ |
| SGDClipMomentum | $0.1220 \pm 0.0162$ | $0.8809 \pm 0.0022$ | $3.6038 \pm 0.0102$ | $59.3052 \pm 0.2798$ |
| Adam | $0.1161 \pm 0.0111$ | $0.8823 \pm 0.0041$ | $3.7692 \pm 0.0062$ | $\mathbf{58.9005 \pm 0.3058}$ |
| Our Algorithm 1 | $\mathbf{0.1086 \pm 0.0129}$ | $\mathbf{0.8835 \pm 0.0032}$ | $3.7928 \pm 0.0425$ | $58.9661 \pm 1.5218$ |

$0.0001$ and the momentum parameter $(\beta_1)$ to be $0.9$. Figure 4 and Table 1 report the training and testing performance for each algorithm, showing that ours is among the best.

**LSTM for Language Modeling on Penn Treebank** We adopt a 3-layer AWD-LSTM [35] to do language modeling on the Penn Treebank (PTB) dataset [33](word level). The mini-batch size is $40$ and we trained each algorithm for 750 epochs. Apart from the hyperparameters we stated above, we further fine-tuned the weight decay value for all algorithms noticing its significant influence on the performance. We choose the set of hyperparameters that give the smallest final validation perplexity. We report the results in Figure 5 and Table 1. It can be seen that we can match the performance of Adam while beating the others.

## 5.2 Transformers Observe $(L_0, L_1)$-smoothness

For Figure 1 which verifies the original form (1) of the $(L_0, L_1)$ condition using the norm, we followed the method in Section H.3 of [56]. Specifically, given $\boldsymbol{x}_t$ and $\boldsymbol{x}_{t+1}$, denote $\boldsymbol{d} := \boldsymbol{x}_{t+1} - \boldsymbol{x}_t$. We estimate the smoothness at $\boldsymbol{x}_t$ by

$$\hat{L}_t = \max_{\gamma \in \{\delta_1, \delta_2, \ldots, \delta_N\}} \frac{\|\nabla F(\boldsymbol{x}_t + \gamma \boldsymbol{d}) - \nabla F(\boldsymbol{x}_t)\|_2}{\|\gamma \boldsymbol{d}\|_2} ,$$

where $\{\delta_1, \delta_2, \ldots, \delta_N\}$ denotes the sample locations and we use $\{\frac{1}{6}, \frac{2}{6}, \frac{3}{6}, \frac{4}{6}, \frac{5}{6}\}$.

For Figure 3 verifying the coordinate-wise version (2) of the $(\boldsymbol{L_0}, \boldsymbol{L_1})$ condition, note that the equation is symmetric in that if we just swap $\boldsymbol{x}$ and $\boldsymbol{y}$ it shall still holds. Thus, during plotting, we compare $\left|\frac{\partial F}{\partial x_j}(\boldsymbol{x}_{t+1}) - \frac{\partial F}{\partial x_j}(\boldsymbol{x}_t)\right|/|x_{t+1,j} - x_{t,j}|$ vs. $\min\left(\left|\frac{\partial F}{\partial x_j}(\boldsymbol{x}_t)\right|, \left|\frac{\partial F}{\partial x_j}(\boldsymbol{x}_{t+1})\right|\right)$.

Figure 1(a) is on training a 2-layer Transformer Encoder to do language modeling on the Wikitext-2 dataset. The implementation, settings, and parameter choices follow this.[1] We only plot the first 5 training epochs. Figure 1(b) and 3 are on training a 6-layer Transformer [50] to do machine translation on the WMT'16 Multimodal Machine Translation Task German-English dataset. The implementation of the transformer is forked from here[2] and we also follow their default settings. The mini-batch size is $256$ and we trained for $400$ epochs using Adam and report the whole training trajectory.

## 5.3 Clipping does not Affect Adam's Performance

We compare clipping and non-clipping for Adam optimizer on the Wikitext-103 (103 million tokens, 180MB) [36] language modeling task, with a 16-layer GPT-2 transformer model [42]. This GPT-2 model has an input length of 256 tokens, 410-dimension word embedding, 16 Attention layers with 10 Attention heads and 2100 hidden dimensions. Model size is 201.58 MB. The vocabulary size is 28996. We use the hyper-parameter settings prescribed in [53]: batch size 256, warm up learning rate from 0 to $2.5 \times 10^{-4}$ in the first 64000 samples (i.e., 250 iterations) and then cosine-anneal learning rate to zero, on top of an Adam optimizer. It takes about 40 hours to train 200 epochs on 8 V100 GPUs. We use clipping threshold max_norm 0.25 for the entire model as prescribed in the literature [53]. We also count that with this clipping scheme, clipping occurs in every single batch. As we can see from Figure 2, neither training loss (2.79 vs 2.76) nor perplexity score (27.92 vs 27.97) differs much in the clipping and non-clipping case, which is consistent with our theory that Adam naturally achieves gradients clipping effect.

---

[1]https://pytorch.org/tutorials/beginner/transformer_tutorial.html
[2]https://github.com/jadore801120/attention-is-all-you-need-pytorch

# 6  Conclusion and Limitations

Smoothness has been a widely adopted condition for proving convergence rates of algorithms in the non-convex optimization scenario. Yet, it has been found that this assumption does not capture losses when employing some deep learning models including RNNs and LSTMs. In light of this, a relaxed smoothness assumption was proposed that aligns well with the practice. We observed that the loss surface of training using Transformers also exhibits this relaxed smoothness. Under this assumption, SGD with clipped gradient has been proven to work well. However, we found that clipping is not necessary for achieving convergence in such a setting. Indeed, we showed that a generalized SignSGD algorithm does not require explicit clipping but can almost guarantee the same bound as SGD with clipping. In the analyses, we identified the key effect of using momentum in analyzing Adam-type algorithms, that it reduces both the noise and the unbounded gradient norms. Finally, we conducted a variety of deep learning tasks showing that our algorithm can match Adam's performance while exceeding others.

**Limitations** The current work is in no way a perfect one and there are many directions worth exploring beyond it. First of all, though our algorithm could be seen as a close resemblance to the original Adam algorithm, they are still not equal. Considering the huge popularity of Adam and its established effectivity in practice, it is worth studying whether Adam in its original form can converge in the relaxed smooth setting. Second, while our Theorem 1 are upper bounds and cannot be directly compared between the two cases of $\beta_2$, it does suggests that $\beta_2 = 0$ minimizes the worst-case convergence rate. However, it still does not fully explain the phenomenon that a choice of $\beta_2$ close to 1 yields better performance in using our Algorithm 1 as well as Adam in practice. Third, despite there are lower bounds showing that, for example, GD with a constant step size can be arbitrarily worse than GD with clipping, it would be more meaningful to study whether the relaxed smooth condition is inherently more difficult, possibly by establishing a lower bound for all first-order optimization algorithms. Fourth, we did show that Transformers observe the relaxed smoothness condition, but we consider it more beneficial to research in-depth what properties or structures make a model satisfy such conditions. Finally, when conducting our experiments, we observed that the weight decay value plays a prominent role in each optimizer's performance, and that the best weight decay value varies for different optimizers. Thus, one potential direction would be to explore different ways of incorporating the regularization in a way to preserve the scale-freeness [37, 38] of Algorithm 1, just as AdamW [31] does [58].

## Acknowledgements

Michael Crawshaw is supported by the Institute for Digital Innovation fellowship from George Mason University. Michael Crawshaw and Mingrui Liu are both supported by a grant from George Mason University. Francesco Orabona is supported by the National Science Foundation under the grants no. 1908111 "AF: Small: Collaborative Research: New Representations for Learning Algorithms and Secure Computation", no. 2022446 "Foundations of Data Science Institute", and no. 2046096 "CAREER: Parameter-free Optimization Algorithms for Machine Learning". The majority of work of Zhenxun Zhuang was done when he was a Ph.D. student at Boston University.

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
