# OpenReview forum: "Robustness to Unbounded Smoothness of Generalized SignSGD"
_NeurIPS.cc/2022/Conference — NeurIPS 2022 Accept_

### Official Review · Reviewer_AWaX · 2022-07-07

**Rating:** 7
**Confidence:** 5
**Soundness:** 3 good
**Presentation:** 3 good
**Contribution:** 3 good

**Summary:**

This paper tackles the problem of optimizing non-convex stochastic functions with relaxed smoothness assumptions, e.g. the $(L_0,L_1)$-smoothness, where the smoothness parameter (i.e. upper bound on the second-order derivative) can be unbounded and locally proportional to the gradient magnitude. Building on prior works [1,2], this paper studies a more fine-grained smoothness, namely the coordinate-wise $(L_0,L_1)$-smoothness. Under this assumption, the authors propose an algorithm based on normalized gradient with momentum and analyze its convergence guarantee. The resulting complexity bound has a similar form to [2], showing that the efficacy of the algorithm does not depend on $L_1$ or the gradient upper bound when $\epsilon$ is small enough. The authors also give experiments to show that coordinate-wise $(L_0,L_1)$-smoothness is reasonable and the proposed algorithm is practically efficient.

[1] Why gradient clipping accelerates training: A theoretical justification for adaptivity. ICLR 2020.

[2] Improved Analysis of Clipping Algorithms for Non-convex Optimization. NeurIPS 2020.

**Questions:**

See the above section.

**Limitations:**

No negative societal impact

**Strengths And Weaknesses:**

**Strengths**:
1. (**Relevance and significance**) Non-convex optimization plays a central role in modern deep learning. Traditionally, the analysis of non-convex optimization algorithms relies heavily on the smoothness of objective functions, in which case vanilla SGD is provably optimal. However, such an assumption is quite unrealistic in practice, especially in training deep neural networks such as LSTMs, for which SGD is much worse than adaptive gradient methods empirically (e.g., Adam). Recently, such an inconsistency between theory and practice has been partially addressed by introducing a relaxed smoothness assumption [1,2]. However, their assumption is still not satisfactory, as the linear dependency between smoothness and gradient norm is very coarse and quite artificial. Investigating better smoothness conditions that align well with practice is clearly of significant value. This paper takes a step towards such a research problem, which in my opinion is important and relevant to the optimization community.

2. (**Writing**). This paper is well-written, clear, and easy to read. The related work section is comprehensive, despite there is an important work missing (see the Weakness part). The comparison between this paper and prior works is well-explained.

3. (**Technical novelty**). I believe studying coordinate-wise smoothness is novel and the proposed algorithm is clean. The similarity of the proposed algorithm with Adam is quite interesting.

**Weaknesses**:

However, I have to say currently this paper have several major weaknesses. I hope they can be addressed before acceptance.

1. (**Missing related work**). A recent work [3] tackled almost the same problem with a similar algorithm as proposed in this paper, and thus should be discussed in detail. In particular, the work of [3] studied the algorithm called normalized SGD with momentum under the $(L_0,L_1)$-smoothness assumption (Algorithm 1 and Theorem 3.5 in their paper). They already pointed out how gradient normalization and momentum can be combined together when dealing with $(L_0,L_1)$-smooth functions with noisy gradients, and how a carefully chosen momentum factor reduces the noise and the required mini-batch size. They also pointed out the relationship with Adam. The only difference is that this paper studies coordinate-wise smoothness and thus uses coordinate-wise gradient normalization. Therefore, I suggest the authors to revise some statements in this paper to avoid overclaim, for example, the abstract/conclusion sections, as well as the following lines:
- Line 74-76: "To the best of our knowledge, this work is the first one establishing guarantees for generalized SignSGD as well as Adam-type updates under a relaxed smoothness condition."
- Line 8: "we show that clipping is not indispensable in tackling such scenarios...... Our analysis underlines the critical role that momentum plays: ...... but it also substantially reduces the effects of unbounded smoothness and gradient norms."

  Moreover, [3] also relaxed the bounded noise assumption to a much weaker assumption, which is better than this paper. They also found a concrete problem called the distributionally robust optimization, which is provably $(L_0,L_1)$-smooth. On the other hand, compared with [3], the strength of this paper lies in the coordinate-wise smoothness / coordinate-wise normalization which are different and novel.

2. (**Regarding the Assumption**) The proposed smoothness assumption seems unreasonable if I understand correctly. For example, consider the quadratic function $F(x)=\frac 1 2 (x_1+x_2)^2$. It does not satisfy the inequality:
\begin{equation}
\left|\frac{\partial F}{\partial x_{j}}(\boldsymbol{y})-\frac{\partial F}{\partial x_{j}}(\boldsymbol{x})\right| \leq\left(L_{0, j}+L_{1, j}\left|\frac{\partial F}{\partial x_{j}}(\boldsymbol{x})\right|\right)\left|y_{j}-x_{j}\right|
\end{equation}
because $\left|\frac{\partial F}{\partial x_{j}}(\boldsymbol{y})-\frac{\partial F}{\partial x_{j}}(\boldsymbol{x})\right|$ can be arbitrarily large as $y_i,i\neq j$ can be chosen arbitrarily which however does not change the right hand side. This seems problematic as even the simple quadratic functions (which are smooth and convex) do not satisfy the condition. The authors should illustrate what functions satisfy this assumption and why it is acceptable.

3. (**Regarding the main results**)
- The main result only provides a complexity upper bound in terms of $\min_t ||\nabla F(x_t)||_1$. This seems a bit weak since $\min_t ||\nabla F(x_t)||_1\le \epsilon$ does not even indicate convergence of the algorithm. It is likely that the gradients near the end of training (say, the last $0.1T$ iterations) are all large. Can you use better metrics to derive the complexity bounds, such as $\frac 1 T \sum_t ||\nabla F(x_t)||_1$ as in [2], or other better forms?
- It seems that $\beta_2=0$ gives the best bound: $\rho$ reaches the maximal value and the algorithm does not require the extra assumption of the gradient upper bound $M_j$. So what is the advantages of increasing $\beta_2$? Why do the authors consider the general case when $\beta_2>0$?
- This paper makes the assumption that the gradient noise is bounded, which is somewhat restrictive (while it is indeed used in prior works [1,2] for analyzing gradient clipping). However, since [3] successfully derived complexity guarantees of the normalized SGD+momentum algorithm under the weaker and more common noise assumption, I think it is likely that this paper can follow the same techniques to relax the noise assumption. Actually, after coarsely reading the proof I found that part of the analysis (e.g., Lemma 11) is similar to the noise reduction analysis in [3]. This weakness is not a major one, but it would be nice to address it.

Nevertheless, I appreciate the motivation of this paper. The current rating is *not final* and I am willing to increase the score if the major concerns (including concerns 1, 2, and the first of 3) are addressed after the author response.

**Miscellaneous minor issues**: In line 194, it should be $\beta_1<1$.

[3] Non-convex Distributionally Robust Optimization: Non-asymptotic Analysis. NeurIPS 2021.

---

> ### Author Response · Authors · 2022-08-02
> **Thank you for your constructive feedback, please see our response. [3/3]**
>
> **Q4: It seems that $\beta_2 = 0$ gives the best bound: $\rho$ reaches the maximal value and the algorithm does not require the extra assumption of the gradient upper bound $M_j$. So what is the advantages of increasing $\beta_2$? Why do the authors consider the general case when $\beta_2 > 0$?**
>
> > A: The reviewer is right in noticing that $\beta_2 = 0$ is the setting minimizing the worst-case convergence rate. However, real-world tasks are rarely worst-case. In fact, in our experiments, $\beta_2 = 0$ does not work very well. Instead, we found that setting $\beta_2$ to be close to $1$ yields the best performance. This motivates the study of the general case when $\beta_2 > 0$.
> >
> > Another motivation is the close resemblance between our algorithm and Adam and the fact that Adam typically employs a $\beta_2$ close to $1$. Therefore, by studying the general case when $\beta_2 > 0$, we took a step forward in understanding why Adam works so well in training LSTMs and Transformers.
> >
> > Moreover, the $\beta_2>0$ setting is also the more challenging one, which requires the novel proof method we propose.
>
> **Q5: This paper makes the assumption that the gradient noise is bounded, which is somewhat restrictive (while it is indeed used in prior works [1,2] for analyzing gradient clipping). However, since [3] successfully derived complexity guarantees of the normalized SGD+momentum algorithm under the weaker and more common noise assumption, I think it is likely that this paper can follow the same techniques to relax the noise assumption. Actually, after coarsely reading the proof I found that part of the analysis (e.g., Lemma 11) is similar to the noise reduction analysis in [3]. This weakness is not a major one, but it would be nice to address it.**
>
> **[3] Non-convex Distributionally Robust Optimization: Non-asymptotic Analysis. NeurIPS 2021.**
>
> > A: Thank you for raising this concern. We do have a remark stating that the bounded gradient noise assumption can be relaxed to a sub-gaussian one (see Section 4.1). We have decided to use the bounded assumption to simplify the proof a bit, which is already extremely technical. Also, we would like to add that our bound holds with high probability, which is stronger than the in-expectation ones in [3]. That said, we would definitely strive to study the convergence under a weaker condition of gradient noises in future work.

---

> > ### Comment · Reviewer_AWaX · 2022-08-03
> > **Thank you for the response.**
> >
> > I would like to thank the authors for their detailed response and careful revision of the paper. Let me give summarization below:
> >
> > Q1: I believe the reference is properly discussed. Minor comment: in line 35, normalized momentum can also recover optimal convergence of SGD and the convergence rate does not depend on L1 (which may be worse a mention).
> >
> > Q2: **It seems that such an assumption is still problematic.** Indeed, when $x_j=y_j$, the right hand size becomes 0 but the left hand size is clearly not 0 if $x_i\neq y_i$ for some $i\neq j$. So the assumption does not hold.
> >
> > Q3: I am satisfactory with the author response. But I may suggest the authors adding a remark to explicitly show that the average-iterate complexity bound can be proved for the case $\beta_2=0$ in the main paper. Some minor comments: while your illustration is correct, the average-iterate implies the convergence of a random iterate which I believe is important, and it is significantly better than the bound of $\min_t ||\nabla F(x_t)||$ (again consider the last $0.1T$ iterates). Also, your counterexample does not actually hold in practice since the parameters for two adjacent updates is small because the learning rate is typically $O(\epsilon)$.
> >
> > Q4: The authors said that "by studying the general case when $\beta_2>0$, we took a step forward in understanding why Adam works so well in training LSTMs and Transformers". However, the theory is not consistent with the experiments and do not explain why in practice one often chooses $\beta_2>0$. While I believe this is a hard problem that may not be addressed in this work, I think the authors should discuss the limitation in the paper.
> >
> > Q5: The authors said that "our bound holds with high probability, which is stronger than the in-expectation ones in [3]." I don't think they are comparable and high probability bound does not indicate an in-expectation bound. Conversely, in-expectation bound implies high probability using concentration bounds, e.g., Markov inequality.
> >
> > Overall, I think currently the main concern lies in the assumption and the current version is still problematic. If the authors can fully address this question, I will increase the score to at least 6.

---

> > > ### Author Response · Authors · 2022-08-06
> > > **Thank you so much for your feedback, we would like to further address your concerns below. [1/2]**
> > >
> > > **Q1: I believe the reference is properly discussed. Minor comment: in line 35, normalized momentum can also recover optimal convergence of SGD and the convergence rate does not depend on L1 (which may be worse a mention).**
> > >
> > > > A: Thanks for noting this, we have added the citation.
> > >
> > > **Q2: It seems that such an assumption is still problematic. Indeed, when $x_j=y_j$, the right hand side becomes 0 but the left hand side is clearly not 0 if $x_i\neq y_i$ for some $i\neq j$. So the assumption does not hold.**
> > >
> > > > A: Thank you so much for observing this! We have slightly changed the assumption (Please see Assumption 2 in the updated version).
> > > >
> > > > The motivations for this change are the following:
> > > > 1. It can now imply the $(L_0, L_1)$ condition using the norm namely Remark 2.3 in [Zhang et al., NeurIPS 2020] when $L_{0, j} = L_0$ and $L_{1, j} = L_1$ for all $j\in[d]$ (up to constants).
> > > >
> > > > 2. Observing Remark 2.3 in [Zhang et al., NeurIPS 2020], we change the left-hand side to a coordinate-wise version thus reducing the scale by a factor of roughly $\sqrt{d}$, then a similar reduction in scale should be expected on the right-hand side. To do this, for the $K_0$ part, we can only change $||x - y||_2$ to $|x_j - y_j|$. Yet, for the $K_1$ part, we can either (1) change $||x - y||_2$ to $|x_j - y_j|$ or (2) change $||\nabla F(x)||_2$ to $|[\nabla F(x)]_j|$, but *not both* (which is what we did in the first version and we now know it was erroneous thanks to your comment). Doing (1) would not work for the quadratic counterexample you raised, but doing (2) can handle this case.
> > > >
> > > > We have carefully gone over our entire proof but it turns out that the needed changes are very minor and the results are very similar (up to some polynomial factors of $d$). We hope to fully solve your concern now. Thank you again for helping us make the paper stronger, we really enjoy communicating with you and appreciate your efforts: this might be the most useful review we received in years!
> > > >
> > > > [Zhang et al., NeurIPS 2020] Bohang Zhang, Jikai Jin, Cong Fang, and Liwei Wang. "Improved analysis of clipping algorithms for non-convex optimization." Advances in Neural Information Processing Systems 33 (2020): 15511-15521.
> > >
> > > **Q3: I am satisfactory with the author response. But I may suggest the authors adding a remark to explicitly show that the average-iterate complexity bound can be proved for the case $\beta_2=0$ in the main paper. Some minor comments: while your illustration is correct, the average-iterate implies the convergence of a random iterate which I believe is important, and it is significantly better than the bound of $\min_t ||\nabla F(x_t)||$ (again consider the last $0.1T$ iterates). Also, your counterexample does not actually hold in practice since the parameters for two adjacent updates is small because the learning rate is typically $O(\epsilon)$.**
> > >
> > > > A: Thank you for your suggestion, we have added this remark (Remark 2, Line 188-189). Also, we agree with your last comment but we would like to point out that this is indeed why we specifically wrote “without additional assumptions” in the response when saying “a convergence guarantee on the average is essentially equivalent to one on the min”. As an example, such a result of the min form is all that can be shown for certain important methods, such as the nonlinear conjugate gradient method. [Theorem 5.7, Nocedal & Wright, 2006]
> > > >
> > > > [Nocedal & Wright, 2006] J. Nocedal and S. J. Wright. Numerical Optimization. Springer New York, Second edition, 2006.
> > >
> > > **Q4: The authors said that "by studying the general case when $\beta_2>0$, we took a step forward in understanding why Adam works so well in training LSTMs and Transformers". However, the theory is not consistent with the experiments and do not explain why in practice one often chooses $\beta_2>0$. While I believe this is a hard problem that may not be addressed in this work, I think the authors should discuss the limitation in the paper.**
> > >
> > > > A: Thank you for your suggestion, we have added this to the Limitations.

---

> > > > ### Comment · Reviewer_AWaX · 2022-08-06
> > > > **Regarding the Assumptions**
> > > >
> > > > I would like to thank the authors again for their efforts in fixing the assumptions. However, sadly, it seems that the current version is **still not satisfactory**. Let me detail the problem below.
> > > >
> > > > Consider the simple quadratic function $f(x)=\frac 1 2 (x_1+x_2)^2$ again. Take $x=(0,0)^T$ and $y=(0,c)^T$ where $c$ is a small constant. As long as $c$ is small enough, the assumption will hold for the two points $x$, $y$. Note that $\nabla f(x)=0$ when $x=(0,0)^T$. Therefore, the assumption implies that
> > > > $$|y_1+y_2|\le L_{0,1} |y_1|$$
> > > > However, $y_1=0$ and $y_2\neq 0$, which yields a contradiction. Thus the current assumption still does not hold for simple quadratic functions.
> > > >
> > > > I am really eager to see the author address this issue, since I appreciate the motivation of this paer and all my other concerns have been suitably addressed.

---

> > > > > ### Author Response · Authors · 2022-08-07
> > > > > **Thank you again for your prompt feedback and we would like to clarify your concerns below:**
> > > > >
> > > > > We have changed the assumption again to admit the counterexample you raised (Please see Assumption 2 in the updated version).
> > > > >
> > > > > 1. It can still imply the $(L_0, L_1)$ condition using the norm namely Remark 2.3 in [Zhang et al., NeurIPS 2020] when $L_{0, j} = L_0$ and $L_{1, j} = L_1$ for all $j\in[d]$ (up to constants).
> > > > >
> > > > > 2. The scales of $L_{0,j}$ and $L_{1,j}$ are still consistent with $L_0$ and $L_1$ of Remark 2.3 in [Zhang et al., NeurIPS 2020] respectively.
> > > > >
> > > > > We have revised our entire proof carefully and again the needed changes are minor and the results are similar.
> > > > >
> > > > > We also discussed in Remark 3 the comparison between our bound and the bound in [Zhang et al., NeurIPS 2020].
> > > > >
> > > > > We hope to have addressed your concern and thank you again for helping us make the paper stronger!
> > > > >
> > > > > [Zhang et al., NeurIPS 2020] Bohang Zhang, Jikai Jin, Cong Fang, and Liwei Wang. "Improved analysis of clipping algorithms for non-convex optimization." Advances in Neural Information Processing Systems 33 (2020): 15511-15521.

---

> > > > > > ### Comment · Reviewer_AWaX · 2022-08-07
> > > > > > **Thank you for your response.**
> > > > > >
> > > > > > It is nice to see that now the assumption is reasonable and holds for quadratic functions. However, I have the following question on which I get confused. Could the author give an explanation to the following question?
> > > > > >
> > > > > > As shown in your paper, Assumption 2 implies the $(L_0,L_1)$-smoothness when $L_{0,j} = L_0$ and $L_{1,j} = L_1$ for all $j\in[d]$, up to constants. In the general case when $L_{0,j}$ or $L_{1,j}$ has different elements, Assumption 2 implies a trivial bound of $(K_0,K_1)$-smoothness for $K_0=O(||L_0||_\infty)$ and $K_1=O(||L_1||_\infty)$. Directly applying the theorem in [Zhang et al., NeurIPS 2020] already gives a complexity of $O(\Delta ||L_0||_\infty ||\sigma||_2 \epsilon^{-4})$ (assuming $\epsilon$ sufficiently small). However, the bound in this paper (Theorem 2) is $O(\Delta ||L_0||_1 ||\sigma||_1 \epsilon^{-4})$ which is strictly larger as $||L_0||_1\ge ||L_0||_\infty$ (despite the theorem is described in terms of high probability). In other words, directly applying [Zhang et al., NeurIPS 2020] without a fine-grained analysis of coordinate-wise smoothness already yields a bound better than this paper. (Please correct me if I understand wrong.)
> > > > > >
> > > > > > If the reason of the above problem still lies in the assumption, I am not sure if it can be addressed by directly proposing the following assumption *based on the proof of descent inequality* (similar to the original assumption proposed by the authors):
> > > > > >
> > > > > > > **Assumption (informal): $F$ is $(L_0,L_1)$-smooth if for any $x$ and $y$ where $||x-y||$ is sufficiently small** (I do not go into the detail which norm $||\cdot||$ is the most natural one and how to define "sufficiently small"), **the following holds:**
> > > > > > $$|\langle \nabla F(y)-\nabla F(x),y-x\rangle|\le \frac 1 2 \left(\sum_j \left(L_{0,j}+L_{1,j}\left|\frac {\partial F}{\partial x_j}(x)\right|\right)|y_i-x_i|^2\right)$$
> > > > > >
> > > > > > - When $L_{1,j}=0$ and $L_{0,j}=L$, the above assumption is a necessary condition to $||\nabla F(y)-\nabla F(x)||\le L||y-x||$, and thus the above condition generalizes standard $L$-smoothness. (It thus holds for quadratic functions.)
> > > > > > - When $L_{1,j}=L_1$ and $L_{0,j}=L_0$, the above assumption can imply the descent inequality for $(L_0,L_1)$-smooth functions.
> > > > > > - The above assumption can lead to a similar descent inequality following your proof, but the $||y-x||_2$ term can be replaced by $|y_j-x_j|$ (which may be better for subsequent proofs).
> > > > > >
> > > > > > In any case, is it possible to prove a bound that can be better than [Zhang et al., NeurIPS 2020] for some specifically chosen $L_0$ and $L_1$ coordinate-wisely?

---

> > > > > > > ### Author Response · Authors · 2022-08-07
> > > > > > > **Thank you for your additional feedback and we clarify your concerns below**
> > > > > > >
> > > > > > > We would like to point out that our bound is in terms of $||\nabla F(x)||_1$ while the bound in [Zhang et al., NeurIPS 2020] is in terms of $||\nabla F(x)||_2$. Hence, we control a bigger quantity (this is better because a bound on our LHS implies a bound on their LHS); yet, the RHS of the inequalities are different and overall these bounds are not directly comparable and it depends on the concrete settings.
> > > > > > >
> > > > > > > As an example of when our bound can be more advantageous, consider the setting when the gradients are dense, then $||\nabla F(x)||_1$ could be in the order of $\sqrt{d}||\nabla F(x)||_2$. Thus, when specifying the precision $\epsilon$ for our algorithm, we actually get $||\nabla F(x)||_1 \leq \epsilon$ which means $\sqrt{d}||\nabla F(x)||_2 \leq \epsilon$. Consequently, for a fair comparison, our algorithm only needs to achieve a precision of $\sqrt{d}\epsilon$ namely $||\nabla F(x)||_1 \leq \sqrt{d}\epsilon$ to guarantee $||\nabla F(x)||_2 \leq \epsilon$. Taking this difference of $\sqrt{d}$ in $\epsilon$ between our bound and theirs into consideration, to guarantee to find a point
> > > > > > > $x$ with $||\nabla F(x)||_2 \leq \epsilon$ and assume $\epsilon$ is sufficiently small, our bound gives a complexity of $O(\Delta||L_0||_1||\sigma||_1^2d^{-2}\epsilon^{-4})$ while their bound gives a complexity of $O(\Delta||L_0||\_{\infty}||\sigma||_2^2\epsilon^{-4})$.
> > > > > > >
> > > > > > > Now, considering that $||L_0||_1 \leq d ||L_0||\_{\infty}$ and that $||\sigma||_1 \leq \sqrt{d}||\sigma||_2$, our bound can match theirs even in the worst case when $L\_{0, j}$ and $\sigma_j$ are the same across $j$, which is intuitive as there is no much difference between dimensions and the original form of $(L_0, L_1)$ is good enough.
> > > > > > >
> > > > > > > With that said, still consider the above setting of gradients being dense, if $||L_0||_1= O(||L_0||\_{\infty})$ and $||\sigma||_1 = O(||\sigma||_2)$ which is the case when, for example, one dimension dominates the others in both $L\_{0,j}$ and $\sigma_j$, then our complexity could be $d^{-2}$ of theirs. This is why we say when $L\_{0,j}$ (and/or $L\_{1,j}$) are different across $j$, we can get finer-grained control.
> > > > > > >
> > > > > > > We also thank you for proposing the alternative condition and have tried it. However, in our proof, we do need a finer-grained condition on the gradient differences, namely for each dimension, and a condition on the inner product is not enough.
> > > > > > >
> > > > > > > We hope this addresses your concern and thank you again for your very constructive feedback.
> > > > > > >
> > > > > > > [Zhang et al., NeurIPS 2020] Bohang Zhang, Jikai Jin, Cong Fang, and Liwei Wang. "Improved analysis of clipping algorithms for non-convex optimization." Advances in Neural Information Processing Systems 33 (2020): 15511-15521.

---

> > > > > > > > ### Comment · Reviewer_AWaX · 2022-08-08
> > > > > > > > **Thank you for your response. All my concerns have been clarified.**
> > > > > > > >
> > > > > > > > I really thank the authors for the detailed response these days, and I believe all my concerns have been clarified. The only minor comment is that the assumption may not need the twice-differentiability of $F$. I have adjusted the score and I believe I can also turn up my confidence level.
> > > > > > > >
> > > > > > > > After the discussion, I may suggest the authors checking the proof again carefully since the proof is quite non-trivial and the assumption has changed for a few times. Also, in the final version the authors may use the extra one page to explain further on some important details during the discussion.

---

> > > > > > > > > ### Author Response · Authors · 2022-08-08
> > > > > > > > > **Thank you!**
> > > > > > > > >
> > > > > > > > > Thank you very much for spending a huge amount of time on our paper. We will definitely incorporate all points during the discussion phase in the final version of our paper.

---

> > > ### Author Response · Authors · 2022-08-06
> > > **Thank you so much for your feedback, we would like to further address your concerns below. [2/2]**
> > >
> > > **Q5: The authors said that "our bound holds with high probability, which is stronger than the in-expectation ones in [3]." I don't think they are comparable and high probability bound does not indicate an in-expectation bound. Conversely, in-expectation bound implies high probability using concentration bounds, e.g., Markov inequality.**
> > >
> > > > A: Thank you for your concern! We acknowledge that the two results in [3] and ours are not directly comparable. Yet, the reason is not between high probability bounds and in-expectation bounds; it is that the assumptions on noise are different. Indeed, the noise assumption in [3] is weaker than ours, so it is expected that [3] does not obtain a high probability bound like ours.
> > > >
> > > > However, we would like to point out that, with a high probability bound *like the one in our Theorem 1*, we know an exponential bound on the tail of the probability distribution, from which computing a bound on the expectation is standard. In contrast, when we only know the expectation of a probability distribution, it is impossible to recover an exponential bound on the tail of the distribution.
> > > >
> > > > Indeed, a high probability bound *like the one in our Theorem 1* always indicates an in-expectation bound which can be done by integrating the tail of the probability distribution. Specifically, our bound says that $Pr[\min\_{t\in[T]} ||\nabla F(x_t)||_1 \geq C\log(dT/\delta)] \leq \delta$ where $C$ contains all other items in the bound. Then take a sequence of $\delta_k$s as $1, \frac12, \frac14, \ldots, \frac{1}{2^k},\ldots$, we know that $Pr[\min\_{t\in[T]} ||\nabla F(x_t)||_1 \geq C\log(2^{k}dT)] \leq \frac{1}{2^k}$.  Thus, we can divide the range of $\min\_{t\in[T]} ||\nabla F(x_t)||_1$ into consecutive intervals $[0, C\log(2^1dT)], [C\log(2^1dT), C\log(2^2dT)], \ldots, [C\log(2^{k-1}dT), C\log(2^{k}dT)], \ldots$ denoted as $I_1, I_2, \ldots, I_k, \ldots$. For each interval $I_k$, the maximum value is $C\log(2^{k}dT)$ and $$Pr[\min\_{t\in[T]} ||\nabla F(x_t)||_1 \in I_k] \leq Pr[\min\_{t\in[T]} ||\nabla F(x_t)||_1 \geq C\log(2^{k-1}dT)] \leq \frac{1}{2^{k-1}} \text{ for } k \geq 2,$$ and note that when $k = 1$, it also holds that $Pr[\min\_{t\in[T]} ||\nabla F(x_t)||_1 \in I_1] \leq 1 = \frac{1}{2^{1-1}}$. Therefore, we can compute the expectation as $$E[\min\_{t\in[T]} ||\nabla F(x_t)||_1] \leq \sum^{\infty}\_{k=1} C\log(2^kdT) \times \frac{1}{2^{k-1}} \leq 2C\log(4dT).$$ As another example, see Lemma A.3 in [Shalev-Shwartz & Ben-David, 2014].
> > > >
> > > > Conversely, while it is true that an in-expectation bound can be transformed into a high probability bound using Markov inequality, this comes with a worse dependency on $\delta$. Specifically, with Markov inequality, we have that $Pr[X \geq a] \leq \frac{E[X]}{a}$. Now consider $\frac{E[X]}{a}$ as $\delta$, the bound is on $Pr[X \geq O(\frac{1}{\delta})]$, which is looser than our bound on $Pr[X \geq O(\log(1/\delta)]$. Indeed, consider when $a = E[X]$, then we can only get that $Pr[X \geq E[X]] \leq 1$ which is loose.
> > > >
> > > > [Shalev-Shwartz & Ben-David, 2014] Shai Shalev-Shwartz and Shai Ben-David. Understanding machine learning: From theory to algorithms. Cambridge university press, 2014.

---

> ### Author Response · Authors · 2022-08-02
> **Thank you for your constructive feedback, please see our response. [2/3]**
>
> **Q2: (Regarding the Assumption) The proposed smoothness assumption seems unreasonable if I understand correctly. For example, consider the quadratic function $F(x)=\frac 1 2 (x_1+x_2)^2$. It does not satisfy the inequality:
> \begin{equation}
> \left|\frac{\partial F}{\partial x_{j}}(\boldsymbol{y})-\frac{\partial F}{\partial x_{j}}(\boldsymbol{x})\right| \leq\left(L_{0, j}+L_{1, j}\left|\frac{\partial F}{\partial x_{j}}(\boldsymbol{x})\right|\right)\left|y_{j}-x_{j}\right|
> \end{equation}
> because $\left|\frac{\partial F}{\partial x_{j}}(\boldsymbol{y})-\frac{\partial F}{\partial x_{j}}(\boldsymbol{x})\right|$ can be arbitrarily large as $y_i,i\neq j$ can be chosen arbitrarily which however does not change the right hand side. This seems problematic as even the simple quadratic functions (which are smooth and convex) do not satisfy the condition. The authors should illustrate what functions satisfy this assumption and why it is acceptable.**
>
> > A: We thank you for raising this concern; yet, our theoretical results already hold for a weaker assumption. First, we would like to clarify that our condition only needs to hold locally when, in all the coordinates $j\in [d]$, the absolute difference between two points is smaller than $1/L_{1,j}$. However, after further investigation inspired by your concern, this locality can be further reduced to when $|x_{1,j} - x_{2,j}| \le 1/||L_1||_1$ for all $j\in[d]$ simultaneously. We have done minimal changes to the paper to use this weaker assumption everywhere. With this assumption, the quadratic function you propose can be shown to satisfy this condition. Thanks again for making our results even stronger!
>
>
> **Q3: The main result only provides a complexity upper bound in terms of $\min_t ||\nabla F(x_t)||_1$. This seems a bit weak since $\min_t ||\nabla F(x_t)||_1 \le \epsilon$ does not even indicate convergence of the algorithm. It is likely that the gradients near the end of training (say, the last $0.1T$ iterations) are all large. Can you use better metrics to derive the complexity bounds, such as $\frac 1 T \sum_t ||\nabla F(x_t)||_1$ as in [2], or other better forms?**
>
> > A: We would like to point out that we can indeed achieve an average-iterate norm upper bound for the case of $\beta_2 = 0$ which recovers the SIGNUM algorithm in [1]. Please see Page 24 Line 666-667 in Appendix A.3 in the updated paper. However, the case when $\beta_2 > 0$ is significantly harder to tackle and we can only get a bound of the minimum of norms over $T$. We used the min form in stating Theorem 1 for consistency.
> >
> > That said, the min form upper bound has been widely used in the community, see, for example, Theorem 2.1 in [2], Theorem 4 in [3], Theorem 4 in [4], Theorem 2.3 in [5], and tens of other similar papers. Indeed, the convergence of the average of the gradients is not a useful measure per se, but it is only used because it implies the convergence of the minimum gradient and the convergence of a random iterate.
> >
> > Last, just to be sure we are on the same page, we would like to point out that a convergence guarantee on the average of the gradient is *not* substantially stronger than the guarantee on the minimum we show. In fact, the convergence of the average of the gradients does *not* imply that the gradients converge to 0, but it only implies the weaker result that there exists a *subsequence* that converges. For example, consider $a_t=\log(t)$ when $t$ is a square number and $a_t=1/\sqrt{t}$ otherwise. It is easy to see that $\frac1T \sum_{t=1}^T a_t$ goes to zero when $T\rightarrow \infty$, yet $a_t$ does not converge to 0 and it is not even bounded! Hence, a convergence guarantee on the average without additional assumptions is essentially equivalent to $\lim_{t\rightarrow\infty} \min_t ||\nabla F(x_t)||_1 = 0$ that is what we prove.
> >
> > [1] Jeremy Bernstein, Yu-Xiang Wang, Kamyar Azizzadenesheli, and Animashree Anandkumar. signSGD: Compressed optimisation for non-convex problems. In International Conference on Machine Learning, pages 560–569. PMLR, 2018.
> >
> > [2] Rachel Ward, Xiaoxia Wu, and Leon Bottou. Adagrad stepsizes: Sharp convergence over nonconvex landscapes. In Proceedings of the Thirty-Sixth International Conference on Machine Learning, pages 6677–6686. PMLR, 2019.
> >
> > [3] Xiaoyu Li and Francesco Orabona. On the convergence of stochastic gradient descent with adaptive stepsizes. In Proceedings of the Twenty-Second International Conference on Artificial Intelligence and Statistics, pages 983–992. PMLR, 2019.
> >
> > [4] Matthew Faw, Isidoros Tziotis, Constantine Caramanis, Aryan Mokhtari, Sanjay Shakkottai, Rachel Ward. The Power of Adaptivity in SGD: Self-Tuning Step Sizes with Unbounded Gradients and Affine Variance. In Proceedings of the Thirty-Fifth Conference on Learning Theory, pages 313-355. PMLR, 2022.
> >
> > [5] Xiaoxia Wu, Rachel Ward, Léon Bottou. WNGrad: Learn the Learning Rate in Gradient Descent. arXiv preprint arXiv:1803.02865.

---

> ### Author Response · Authors · 2022-08-02
> **Thank you for your constructive feedback, please see our response. [1/3]**
>
> **Q1: (Missing related work). A recent work [3] tackled almost the same problem with a similar algorithm as proposed in this paper, and thus should be discussed in detail. In particular, the work of [3] studied the algorithm called normalized SGD with momentum under the (L0, L1)-smoothness assumption (Algorithm 1 and Theorem 3.5 in their paper). They already pointed out how gradient normalization and momentum can be combined together when dealing with (L0, L1)-smooth functions with noisy gradients, and how a carefully chosen momentum factor reduces the noise and the required mini-batch size. They also pointed out the relationship with Adam. The only difference is that this paper studies coordinate-wise smoothness and thus uses coordinate-wise gradient normalization. Therefore, I suggest the authors to revise some statements in this paper to avoid overclaim, for example, the abstract/conclusion sections, as well as the following lines:**
>
>   * **Line 74-76: "To the best of our knowledge, this work is the first one establishing guarantees for generalized SignSGD as well as Adam-type updates under a relaxed smoothness condition."**
>
>   * **Line 8: "we show that clipping is not indispensable in tackling such scenarios...... Our analysis underlines the critical role that momentum plays: ...... but it also substantially reduces the effects of unbounded smoothness and gradient norms."**
>
> **Moreover, [3] also relaxed the bounded noise assumption to a much weaker assumption, which is better than this paper. They also found a concrete problem called the distributionally robust optimization, which is provably -smooth. On the other hand, compared with [3], the strength of this paper lies in the coordinate-wise smoothness / coordinate-wise normalization which are different and novel.**
>
> **[3] Non-convex Distributionally Robust Optimization: Non-asymptotic Analysis. NeurIPS 2021.**
>
> > A: Thank you for pointing out this valuable reference, appreciate it! We have revised the paper to include a discussion of this paper in Section 2, sharpened our message on the fact that we analyze Adam-type updates, and added this algorithm in the experimental evaluation.
> >
> > In detail, we do agree that they showed that gradient normalization and momentum can tackle non-smoothness and control noise and that they speculated a relation between the algorithm they proposed and Adam.
> >
> > Nevertheless, that algorithm normalizes each coordinate with the same scale, namely the norm of the momentum. In contrast, Adam operates coordinate-wisely and scales each coordinate differently with second-moment information of past gradients. Moreover, we also observed in experiments that normalized SGD with momentum performs inferior to Adam (please see the updated results in Section 5.1). Thus, we believe that there is still a large gap between the normalized SGD with momentum and Adam, possibly much larger than between our algorithm and Adam.
> >
> > Indeed, as the reviewer pointed out, the strength of our paper “lies in the coordinate-wise smoothness / coordinate-wise normalization which are different and novel” by which our algorithm can closely resemble Adam, both in the shape of updates and in the empirical performance. The Adam-type updates also make the analysis substantially more difficult and they require the novel proof method we introduced which is not in [3]. Hence, we believe our contribution is significant and lays out a very potential way towards understanding Adam’s performance in deep learning tasks, especially in training LSTMs and Transformers.

---

### Official Review · Reviewer_3xib · 2022-07-13

**Rating:** 5
**Confidence:** 4
**Soundness:** 3 good
**Presentation:** 3 good
**Contribution:** 3 good

**Summary:**

In this paper, the authors firstly studied the applicability and generalization of the relaxed smoothness assumption. Empirically, the authors showed that Transformer, one of the most widely used practical models, also satisfied the relaxed smoothness assumption for some $L_0, L_1$, which showed the assumption is meaningful and not too strong.  Secondly, given the relaxed smoothness assumption, the authors proposed and analyzed a generalized SignSGD algorithm. From theoretical perspective, the authors not only proved the convergence rates, but also located the critical role of the momentum players: it reduces the noise effect and gives an exponential decaying effect on the unbounded gradient norms and smoothness.

**Questions:**

Is it possible to propose a last-iterate norm upper bound or average-iterate norm upper bound?
By the way, the numbering of the lemmas is quite strange. Lemma 11 is in front of Lemma 9. Could you explain that?

**Limitations:**

I don't think this paper has any potential negative societal impact since it is completely a theory-based work. All the experiments are used to verify their theory and compare with other optimizers.

**Strengths And Weaknesses:**

This paper has high writing quality, complete literature review. The topic is OK in terms of originality and significance. However, according to the authors' experimental results, the generalized SignSGD seems to be a little worse than the best existing algorithm, which may make it difficult to be widely used in practice. On the other hand, in the main theorem, the left hand side is the minimum of norm over T, which I feel quite strange, since the most metric I saw is the last-iterate norm and the average norm. It would be better if the authors can make a clear explanation on this.

---

> ### Author Response · Authors · 2022-08-02
> **Thank you for your constructive feedback, please see our response.**
>
> **Q1: However, according to the authors' experimental results, the generalized SignSGD seems to be a little worse than the best existing algorithm, which may make it difficult to be widely used in practice.**
>
> > A: From our experiment results, ours is either the best one in testing, or the difference between it and the best one (Adam) is very small (for example, in the Penn Treebank experiment, the testing perplexity of $58.9661$ by ours vs. $58.9005$ by Adam, a difference of only $0.066$). Also, given the close resemblance between our algorithm and Adam, both in the updates and the training/testing curves, we consider our theoretical results as very strong candidates to explain the performance of Adam under assumptions that match the reality of deep learning applications.
>
> **Q2: On the other hand, in the main theorem, the left hand side is the minimum of norm over T, which I feel quite strange, since the most metric I saw is the last-iterate norm and the average norm. It would be better if the authors can make a clear explanation on this. Is it possible to propose a last-iterate norm upper bound or average-iterate norm upper bound?**
>
> > A: We would like to point out that we can indeed achieve an average-iterate norm upper bound for the case of $\beta_2 = 0$ which recovers the SIGNUM algorithm in [1]. Please see the end of Proof of Theorem 1 for $\beta_2 = 0$ (Page 24 Line 672-673 in Appendix A.3) in the updated paper. However, the case when $\beta_2 > 0$ is significantly harder to tackle and we can only get a bound of the minimum of norms over $T$. We used the min form in stating Theorem 1 for consistency.
> >
> > That said, the min form upper bound has been widely used in the community, see, for example, Theorem 2.1 in [2], Theorem 4 in [3], Theorem 4 in [4], Theorem 2.3 in [5], and tens of other similar papers. Indeed, the convergence of the average of the gradients is not a useful measure per se, but it is only used because it implies the convergence of the minimum gradient and the convergence of a random iterate.
> >
> > Last, just to be sure we are on the same page, we would like to point out that a convergence guarantee on the average of the gradient is *not* substantially stronger than the guarantee on the minimum we show. In fact, the convergence of the average of the gradients does *not* imply that the gradients converge to 0, but it only implies the weaker result that there exists a *subsequence* that converges. For example, consider $a_t=\log(t)$ when $t$ is a square number and $a_t=1/\sqrt{t}$ otherwise. It is easy to see that $\frac1T \sum_{t=1}^T a_t$ goes to zero when $T\rightarrow \infty$, yet $a_t$ does not converge to 0 and it is not even bounded! Hence, a convergence guarantee on the average, without additional assumptions, is essentially equivalent to $\lim_{t\rightarrow\infty} \min_t ||\nabla F(x_t)||_1 = 0$ which is what we prove.
> >
> > [1] Jeremy Bernstein, Yu-Xiang Wang, Kamyar Azizzadenesheli, and Animashree Anandkumar. signSGD: Compressed optimisation for non-convex problems. In International Conference on Machine Learning, pages 560–569. PMLR, 2018.
> >
> > [2] Rachel Ward, Xiaoxia Wu, and Leon Bottou. Adagrad stepsizes: Sharp convergence over nonconvex landscapes. In Proceedings of the Thirty-Sixth International Conference on Machine Learning, pages 6677–6686. PMLR, 2019.
> >
> > [3] Xiaoyu Li and Francesco Orabona. On the convergence of stochastic gradient descent with adaptive stepsizes. In Proceedings of the Twenty-Second International Conference on Artificial Intelligence and Statistics, pages 983–992. PMLR, 2019.
> >
> > [4] Matthew Faw, Isidoros Tziotis, Constantine Caramanis, Aryan Mokhtari, Sanjay Shakkottai, Rachel Ward. The Power of Adaptivity in SGD: Self-Tuning Step Sizes with Unbounded Gradients and Affine Variance. In Proceedings of the Thirty-Fifth Conference on Learning Theory, pages 313-355. PMLR, 2022.
> >
> > [5] Xiaoxia Wu, Rachel Ward, Léon Bottou. WNGrad: Learn the Learning Rate in Gradient Descent. arXiv preprint arXiv:1803.02865.
>
> **Q3: By the way, the numbering of the lemmas is quite strange. Lemma 11 is in front of Lemma 9. Could you explain that?**
>
> > A: Thank you for noticing this oddity, we have reordered them in the updated version of the paper.

---

> ### Author Response · Authors · 2022-08-08
> **Look forward to post-rebuttal feedback!**
>
> Dear Reviewer,
>
> Thank you so much for spending the time to review our paper. We have carefully answered your questions, including the comparison between our algorithm’s empirical performance and others, the relationship between an average-form bound and our min-form bound, and the numbering of lemmas.
>
> Please let us know if our replies address your concerns and we are glad to discuss any additional questions you may have. With that said, if our response resolves your concerns, we politely invite you to consider raising the rating of our work.

---

### Official Review · Reviewer_JfA2 · 2022-07-17

**Rating:** 6
**Confidence:** 4
**Soundness:** 3 good
**Presentation:** 2 fair
**Contribution:** 2 fair

**Summary:**

This work extends adaptive (L0, L1)-smooth [1] from gradient clip that clips large gradients to signSGD that shrinks the learning rate when the gradient norm is large. The author proves the convergence and shows the experiments.

The core idea is that the gradient smoothness is small (it changes quickly) when the gradient norm is large. Both gradient clip and signSGD reduces the learning rate when the curvature changes too fast.

[1] Zhang, Jingzhao, Tianxing He, Suvrit Sra, and Ali Jadbabaie. "Why gradient clipping accelerates training: A theoretical justification for adaptivity." arXiv preprint arXiv:1905.11881 (2019).

**Questions:**

no

**Limitations:**

see strengths and weaknesses.

**Strengths And Weaknesses:**

Strengths:
1） Based on the (L0, L1)-smooth observation, the extension from gradient clip to signSGD follows the intuition.
2） both theoretical proofs and experimental results are shown.


Weaknesses:
1) The (L0, L1)-smooth observation is well demonstrated by [1].
2) [1] already prove the fast convergence rate of normalized gradient methods with the (L0, L2)-smooth.
The signSGD is close related to the normalized gradient method. (comparing equation (6) in [1] with algorithm 1 line 5-7).
It reduces the novelty is this work.

[1] Zhang, Jingzhao, Tianxing He, Suvrit Sra, and Ali Jadbabaie. "Why gradient clipping accelerates training: A theoretical justification for adaptivity." arXiv preprint arXiv:1905.11881 (2019).

---

> ### Author Response · Authors · 2022-08-02
> **Thank you for your constructive feedback, please see our response.**
>
> **Q1: The (L0, L1)-smooth observation is well demonstrated by [1].**
>
> > A: We acknowledge that the (L0, L1)-smooth condition was first proposed by [1]. Yet, their observation was on training LSTMs only and their form works with norms of gradients and Hessians. As pointed out by Reviewer AWaX, this “assumption is still not satisfactory…Investigating better smoothness conditions that align well with practice is clearly of significant value”.
> >
> > In this paper, *for the first time*, we observed that this extended smoothness condition can nicely characterize the training of Transformers as well (Figure 1). Furthermore, we empirically observed that the original form does not capture the variation across layers/coordinates very well (Figure 3), and thus proposed the coordinate-wise (L0, L1)-smooth condition. Given the increasing popularity of Transformers recently, we consider our contribution non-trivial and significant.
>
> **Q2: [1] already prove the fast convergence rate of normalized gradient methods with the (L0, L1)-smooth. The signSGD is close related to the normalized gradient method. (comparing equation (6) in [1] with algorithm 1 line 5-7). It reduces the novelty of this work.**
>
> > A: The reviewer is right that SignSGD is closely related to the normalized gradient method. However, our main contribution and the major technical difficulty lies in analyzing an entire family of algorithms. Indeed, our generalized SignSGD algorithm recovers the SignSGD when $\beta_1 = \beta_2 = 0$. Even in that case, the normalized gradient method scales each coordinate by a single scalar, namely the norm of the gradient; whereas SignSGD scales each coordinate separately so that they all have an absolute value of 1. As Reviewer AWaX pointed out, “the strength of this paper lies in the coordinate-wise smoothness / coordinate-wise normalization which are different and novel”.
> >
> > Moreover, when $\beta_1 = \beta_2 = 0$ no longer holds, our algorithm becomes less similar to the normalized gradient method. In particular, when both $\beta_1$ and $\beta_2$ are close to 1, it behaves more like Adam (both in the algorithmic form and in practice). Analyzing this case is highly non-trivial and requires the novel proof method we introduce.
> >
> > Furthermore, in doing experiments, we observed that SignSGD with $\beta_1 = \beta_2 = 0$ and the normalized gradient method (new experiments added in the revision) both perform worse than Adam, while setting both $\beta_1$ and $\beta_2$ to be close to 1 yields great results matching that of Adam and beating others. This supports our extension beyond SignSGD and normalized gradient methods. Indeed, our algorithm is a general one that unifies SignSGD and (almost) Adam with a convergence rate strictly better than gradient descent (with a constant step size) and achieves great performance in practice. With that said, it also provides an angle into understanding why Adam works so well in training LSTMs and Transformers.
>
> [1] Zhang, Jingzhao, Tianxing He, Suvrit Sra, and Ali Jadbabaie. "Why gradient clipping accelerates training: A theoretical justification for adaptivity." arXiv preprint arXiv:1905.11881 (2019).

---

> ### Author Response · Authors · 2022-08-08
> **Look forward to post-rebuttal feedback!**
>
> Dear Reviewer,
>
> Thank you so much for spending the time to review our paper. We have carefully answered your questions, including clarifying our paper’s contribution compared with [Zhang et al., ICLR 2020]  and normalized gradient methods.
>
> Please let us know if our replies address your concerns and we are glad to discuss any additional questions you may have. With that said, if our response resolves your concerns, we politely invite you to consider raising the rating of our work.
>
> [Zhang et al., ICLR 2020] Jingzhao Zhang, Tianxing He, Suvrit Sra, and Ali Jadbabaie. Why gradient clipping accelerates training: A theoretical justification for adaptivity. In International Conference on Learning Representations, 2020.

---

### Author Response · Authors · 2022-08-08
**Summary of changes**

Dear Reviewers:

We appreciate your constructive reviews and would like to thank you for your time and efforts. As the deadline of the discussion period is coming close, we would like to summarize the changes we made to the paper in the past two weeks. These changes are inspired by the valuable comments from all reviewers together with a very fruitful discussion with reviewer AWaX and we are confident that our paper is stronger now. Thus, we would like to invite you to re-evaluate our work:

1. We have changed our assumption to make it more practical, hold for more general settings, and be able to imply the original $(L_0, L_1)$ smoothness condition (Remark 2.3 in [Zhang et al., NeurIPS 2020]). We have also gone over the entire proof to make modifications accordingly which turns out to be minor.

2. We have compared our complexity bound with that of [Zhang et al., NeurIPS 2020] and explained when our algorithm could be more advantageous.

3. We have discussed [Jin et al., NeurIPS 2021] in detail, sharpened our message on the fact that we analyze Adam-type updates, and clarified our contribution compared with it: we proposed the *coordinate-wise* $(L_0, L_1)$ smoothness condition inspired by our empirical observation on training Transformers. We then designed a generalized SignSGD algorithm with coordinate-wise normalization that closely resembles Adam, both in the shape of updates and in the empirical performance, and thus lays out a very potential way towards understanding Adam’s performance in deep learning tasks, especially in training LSTMs and Transformers.

4. Considering the relationship between our algorithm and the normalized SGD with Momentum algorithm from [Jin et al., NeurIPS 2021], we added the comparison with it in all experiments and the results show that our algorithm’s performance still exceeds theirs.

[Jin et al., NeurIPS 2021] Jikai Jin, Bohang Zhang, Haiyang Wang, and Liwei Wang. "Non-convex distributionally robust optimization: Non-asymptotic analysis." Advances in Neural Information Processing Systems 34 (2021): 2771-2782.

[Zhang et al., NeurIPS 2020] Bohang Zhang, Jikai Jin, Cong Fang, and Liwei Wang. "Improved analysis of clipping algorithms for non-convex optimization." Advances in Neural Information Processing Systems 33 (2020): 15511-15521.

---

### Meta-Review · Area_Chair_CZVB · 2022-08-29

**Recommendation:** Accept
**Confidence:** Certain

**Metareview:**

During the discussion it became clear that this work naturally extends previous works
on signSGD to encompass generalised methods. Several clarifications were made by the authors throughout the rebuttal process, and this has mostly satisfied the reviewers. Moreover, I think that all in all this work may be interesting to the neurips community, and I recommend to accept it.

**Award:**

No

---

### Decision · Program_Chairs · 2022-09-14

Accept